

# Paranoid thinking and perceived competitive intention

Yutaka Horita

Department of Psychology, Teikyo University, Tokyo, Japan

## ABSTRACT

Paranoid thinking, that others are hostile, can be seen even in the general population. Paranoia is considered the expectation that others are competitors who aim to maximize the differences in payoffs rather than maximize their own payoffs. This study examined whether paranoia reflects the irrational belief that others have a competitive intention and is associated with avoiding perceived competition. We recruited 884 US residents *via* the Internet and conducted a modified Dictator Game, in which monetary allocation was carried out between the Dictator and the Recipient. The Dictator chooses either fair or competitive allocation while selecting the competitive allocation is irrelevant to increasing the Dictator's payoffs. The Recipient decides whether to accept the Dictator's decision or receive sure but low rewards. We found that Recipients with high-level paranoid thinking expected their opponent to select competitive allocation more than those with low levels, even when selecting it was costly for Dictators. Paranoid thinking was not associated with selecting sure rewards or competitive allocations. The results suggest that paranoia reflects the belief that others have a competitive intention but is not related to avoidance behavior against perceived threats and unilateral attacks.

## INTRODUCTION

Paranoia is an unfounded concern that others have intentions to harm (*Freeman & Garety, 2000*). Paranoia reflects mistrust and suspicion of persecution (*Bebbington et al., 2013*), and several subtypes of paranoia have been identified (*Trower & Chadwick, 1995*; *Combs et al., 2007a*). While paranoid thinking is a common symptom of psychotic disorders, it has been observed as a spectrum ranging from low to severe levels of persecutory delusions in the general population (*Freeman et al., 2005*; *Freeman et al., 2011*; *Bebbington et al., 2013*; *Freeman, 2016*; *Elahi et al., 2017*; *Bell & O'Driscoll, 2018*). The cognitive mechanisms of paranoia, including its neurobiological underpinnings (*Pinkham et al., 2015*; *Barnby et al., 2020a*), have been studied in diverse disciplines. From an evolutionary perspective, recent empirical studies have focused on paranoia as a general psychological mechanism for detecting social threats (*Gilbert, 2001*; *Green & Phillips, 2004*; *Raihani & Bell, 2019*; *Bell, Raihani & Wilkinson, 2021*).

Paranoia represents a cognitive bias that attributes the intentions of others as malevolent, even under ambiguous situations (*Garety & Freeman, 1999*; *Murphy et al., 2018*; *Trotta et al., 2021*). People with higher paranoid levels have a cognitive tendency to

Corresponding author
Yutaka Horita,
horita@main.teikyo-u.ac.jp

overattribute others' intentions as malicious during ambiguous hypothetical situations (*Combs et al., 2007b*; *Jack & Egan, 2016*; *So et al., 2020*). Previous studies using incentivized economic games have indicated that a characteristic of paranoia is the attribution or supposition of harmful intentions in social interactions (*Raihani & Bell, 2017*; *Raihani & Bell, 2018*; *Saalfeld et al., 2018*; *Greenburgh, Bell & Raihani, 2019*; *Barnby et al., 2020b*; *Raihani et al., 2021*; *Horita, 2021*). For instance, the Dictator Game (DG) (*Forsythe et al., 1994*) has been conducted to test the influence of paranoia on the attribution of harmful intentions. In this game, the Dictator decides how to distribute money between themselves and the Recipients. Dictators' motivations for unfair allocations are ambiguous: they can be attributed to their harmful intent (*e.g.*, desire to reduce Recipients' payoff) or self-interest (*e.g.*, desire to increase their own payoff). Previous studies have demonstrated that recipients with higher paranoid thinking tend to attribute unfair Dictator's decisions to harmful intent, although the attribution of self-interest does not differ between paranoia levels (*Raihani & Bell, 2017*; *Saalfeld et al., 2018*; *Barnby et al., 2020b*).

It is expected that the concept of "social value orientation" (SVO) (*Messick & McClintock, 1968*) plays a role in understanding the relationship between paranoia and social cognition about others' harmful intent. Similar to the concept of social preferences (*Fehr & Fischbacher, 2002*), SVO describes how individuals weigh the payoffs between themselves and others in resource distributions. Although various types of SVOs can be assumed theoretically, generally, three types of SVOs have been considered in practice: prosocial, individualist, or competitor (*Messick & McClintock, 1968*; *Van Lange et al., 1997*; *Van Lange, 1999*). For example, to measure an individual's SVO, they were presented with different options for resource allocation (as shown in Fig. 1A). The individuals were classified into three orientations according to their allocation preferences. Prosocial refers to the tendency to maximize the joint outcomes of the self and the other, or to achieve equality. Both individualists and competitors reflect proself orientations, where they weigh their own gains more than those of others. However, while individualists maximize absolute outcomes for the self regardless of the other's outcomes, competitors maximize the relative difference in payoffs between themselves and the other.

Recent studies have provided evidence that paranoia is the tendency to infer the intentions of others as harmful rather than self-interested (*Raihani & Bell, 2017*; *Raihani & Bell, 2018*; *Saalfeld et al., 2018*; *Greenburgh, Bell & Raihani, 2019*; *Barnby et al., 2020b*; *Raihani et al., 2021*). In other words, paranoia is associated with the expectation that others possess competitive orientations rather than individualistic ones. However, compared with individualists, competitors are irrational in terms of payoff maximization. In fact, meta-analyses of SVOs indicated that people classified as competitors were rare compared to other SVOs (prosocial, 46%; individualist, 38%; competitor, 12%: *Au & Kwong, 2004*). If we believe that others are rational people who aim to maximize their profits, there is no reason to assume that they will act as competitors.

Previous studies have used DG to examine the connection between paranoid thinking and attribution of the Dictator's selfish behavior (*Raihani & Bell, 2017*; *Saalfeld et al., 2018*; *Barnby et al., 2020b*). However, it is not apparent whether the Dictator's unfair allocations are caused by an individualistic or a competitive orientation in the usual DG. Instead of the

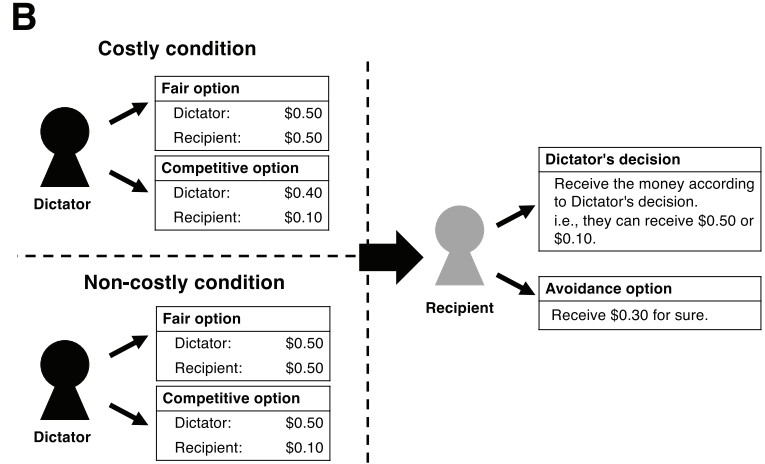

**Figure 1 Schematics of the present study.** (A) Example of three types of SVO. The table shows the distributions that reflect each kind of SVO (prosocial, individualist, and competitor) (*Van Lange et al., 1997*). The number in each cell represents the points that the self or the other earn. (B) Experimental design in the present study. The Dictator chooses either a fair or a competitive option. The Recipient chooses either a Dictator's decision or an avoidance option without being informed of the Dictator's decision.

DG, a previous study (*Horita, 2021*) conducted a Preemptive Strike Game (PSG: *Simunovic, Mifune & Yamagishi, 2013*) to examine how paranoid thinking affected the supposition of others' competitive intentions and responses against the subjective perception of harm in an uncertain situation where the other's behavior was unknown. In the PSG, two players initially receive money as an endowment (*e.g.*, $0.5) and decide whether to attack their partner or not. If neither of them chooses to attack during the allotted time, both receive the initial money (*i.e.*, $0.5). If one of them attacks the partner first, they must pay a small cost (*e.g.*, the attacking player loses $0.1 and earns $0.4). In contrast, the attacked partner loses a larger amount of money (*e.g.*, the attacked player loses $0.4 and earns $0.1). Attacks in the PSG are against maximizing their own payoffs; therefore, there is no reason for people with an individualistic orientation to engage in an attack. People who believe that others are rational would not choose to attack them. However, preemptive attacks can occur driven by the fear that others may have a competitive orientation. Indeed, the PSG experiment observed people who engaged in preemptive attacks because they were driven by fear (*Simunovic, Mifune & Yamagishi, 2013*). A previous study (*Horita, 2021*) showed that in the PSG, participants with higher paranoid thinking assumed that others were driven by harmful intent compared to those

with lower paranoid thinking. However, the effect of paranoia on enhancing preemptive attacks was weak and statistically insignificant. Moreover, the paranoid tendency was correlated with self-reported aggressive motivation (*i.e.*, the desire to reduce the opponent's payoffs).

The purpose of the present study is to examine in detail whether paranoia reflects the belief that others are competitors and is associated with behavior that avoids perceived competitive intention. Although the motivations behind attacking behavior in PSG have been demonstrated to be mainly driven by defensive aggression (*Simunovic, Mifune & Yamagishi, 2013*), logically, both defensive and offensive motivations can be mixed. A previous study also found a correlation between paranoid thinking and self-reported aggressive motivation in PSG (*Horita, 2021*). Another study suggested an association between paranoia and enjoyment of negative social interactions (*Raihani et al., 2021*). Therefore, instead of PSG, we used the following modified DG to exclude the possibility that offensive motivation would work.

In our DG, the Dictator determines how to allocate money between their partners and themselves by choosing one of two options. One option is a fair distribution (*e.g.*, both the Dictator and the Recipient receive $0.5), and the other is a competitive one. According to the options used in the method for classifying SVOs (*Van Lange et al., 1997*), we set two conditions: non-costly and costly. In the non-costly condition, the Dictator's earnings do not change even if they choose the competitive option (*e.g.*, the Dictator receives $0.5, while the Recipient receives $0.1). In the costly condition, if the Dictator chooses the competitive option, their earnings become less than those when choosing the fair option (*e.g.*, the Dictator receives $0.4, while the Recipient receives $0.1). In both conditions, for Dictators, choosing the competitive option is irrelevant to maximizing their payoff. After the Dictator has made their decision, the Recipient decides whether to receive the money according to the Dictator's decision or choose an avoidance option. If the Recipient chooses the avoidance option, they can receive a fixed amount of money (*e.g.*, $0.3). This amount is higher than when the Dictator has chosen the competitive option but lower than when they have chosen the fair option (Fig. 1B depicts the structure of our DG).

In the non-costly condition, choosing the avoidance option would reflect the belief that others were competitors rather than prosocials who cared about the interests of others. Furthermore, the costly condition is a situation in which choosing a competitive distribution is inconsistent with maximizing self-interest. Therefore, in the costly condition, selecting the avoidance option would imply a strong belief that others' orientations were competitive rather than individualistic. Thus, participants who believe that general people behave as competitors may choose the avoidance option in this game, and such a belief may be observed among those with higher paranoid thinking. A questionnaire survey indicated an association between persecution delusions and self-reported experiences of avoiding threats (*Freeman et al., 2007*). However, recent experimental studies using economic games have reported a weak or no association between paranoia and behavioral tendencies to avoid perceived harm (*Greenburgh et al., 2021*; *Horita, 2021*). In contrast to the experimental games used in these previous studies (the PSG in *Horita (2021)* and the trust game in *Greenburgh et al. (2021)*), the DG in the

present study eliminates the possibility that Recipients influence their opponents' payoffs (*e.g.*, reducing their partner's earnings). Moreover, our DG reflects a situation in which the reasons for the Dictator's unfair behavior are more likely to be attributed to competitive orientation rather than individualism. By observing the behavior of the Recipients, we expect to rigorously examine the link between the supposition of competitiveness and avoidance of perceived harm in paranoia.

We examined the following pre-registered predictions concerning the Recipients' behavior (https://osf.io/9s5tf/?view_only=9c62cf2b64324b47a48869a94fcbd8ea):

Prediction 1: highly paranoid people are more likely to expect that others will choose a competitive allocation than less paranoid people.

Prediction 2: highly paranoid people are more likely to suppose that others will have a harmful intent than less paranoid people.

Prediction 3: highly paranoid people are more likely to prefer a reward that they can get for sure to one allocated by others than less paranoid people.

## MATERIALS AND METHODS

This study was conducted between October and November 2021. We recruited participants using Amazon Mechanical Turk (MTurk; www.mturk.com). For recruitment, we used the MTurk toolkit and randomly recruited participants from the "CloudResearch-Approved Participants" pool provided by CloudResearch (www.cloudresearch.com) (*Litman, Robinson & Abberbock, 2017*). This pool consists of MTurk workers who passed attention checks by CloudResearch. We restricted the qualifications for participation to avoid experienced workers (*Litman & Robinson, 2020*): the number of tasks approved by employers was up to 5,000, with an approval rate of more than 90%. In addition to these qualifications, we targeted participants who resided in the U.S. and did not set any other demographic criteria. This study was pre-registered at https://osf.io/9s5tf/?view_only=9c62cf2b64324b47a48869a94fcbd8ea.

### Participants

We recruited participants for a pre-survey to measure their paranoia scores. Seven days after completing the data collection for paranoia scores, we invited the same participants to participate in the DG. After sending the invitations, we stopped data collection when the number of participants in the DG dropped below two per day, according to the pre-registered plan to collect the data. In the DG, we set the conditions as between-subject factors, and participants were randomly assigned to either the costly or non-costly condition. Questionnaire forms for the pre-survey and the DG were developed using Qualtrics (www.qualtrics.com).

First, we collected all data on the Dictators' behavior. Subsequently, we recruited participants who took part in the DG as Recipients. We aimed to include at least 300 participants for each role to conduct robust analyses. Thus, we initially recruited 650 participants within each role for the pre-survey, predicting that the number of candidates for the DG would be reduced. Sample sizes were also determined based on budget constraints.

Finally, we collected a total of 884 participants (563 females, 312 males, and nine answered "prefer not to say"). The mean age of the participants was 39.72 (*SD* = 13.04) years, ranging from 18 to 78 years. We gathered 478 Dictators; 243 were assigned to costly conditions and 235 to non-costly conditions. We gathered 406 Recipients; 206 were assigned to costly conditions and 200 to non-costly conditions.

## Pre-survey

Participants completed the revised version of Green et al.'s Paranoid Thoughts Scale (R-GPTS) (*Freeman et al., 2021*). The R-GPTS consists of a social reference subscale (eight items) and a persecution subscale (10 items). Participants rated each item on a five-point scale ranging from 0 (not at all) to 4 (totally). We used the Persecution subscale for the analyses, as described by *Freeman et al. (2021)*. We summed the scores of each item for each participant; they can range from 0 to 40, with a higher score representing a greater degree of paranoid thinking (hereafter, we refer to this as "paranoia score").

After completing the R-GPTS, participants were asked about their gender ("male," "female," or "prefer not to say") and age. Each participant was paid 0.60 United States dollars (USD) as a reward for participating in the pre-survey.

## Dictator game

Participants were paid 0.60 USD as remuneration to participate in the experiment. Additionally, they could receive earnings in the DG as bonuses. The instructions for the DG are provided in the Supplemental Materials.

First, the participants were provided with instructions for the DG. In the instruction, the Dictator and the Recipient were referred to as "Player 1" and "Player 2," respectively. Participants were told that they were randomly paired with another participant and assigned to either the Dictator or the Recipient. Money was allocated only once between the two.

The Dictator decided the allocation by selecting one of the two options that an experimenter prepared. Referring to "decomposed games" used for measuring SVO (*Van Lange et al., 1997*), we presented two allocation options to Dictators: a "fair" or a "competitive" option. These options are referred to as A and B, respectively, in the instructions. In both the costly and non-costly conditions, the fair option was set to "0.50 USD to Dictator, and 0.50 USD to Recipient." In the costly condition, the competitive option was set to "0.40 USD to Dictator, and 0.10 USD to Recipient;" the Dictators must sacrifice 0.10 USD if they choose the competitive option. In the non-costly condition, the competitive option was set to "0.50 USD to Dictator, and 0.10 USD to Recipient;" the earnings of Dictators did not change whether they had chosen the fair or competitive option. In both conditions, the Recipients' earnings were set to 0.10 USD when the Dictator chose the competitive option. Dictators were not informed of the Recipients' decision-making. Dictators received money that they distributed as bonuses.

The Recipients were informed of the two allocation options presented to the Dictator according to their assigned condition and that they took their decisions by selecting either the fair or competitive options. Recipients decided to choose either "to receive the money

according to the Dictator's decision" (referred to as option X) or "to receive 0.30 USD for sure" (*i.e.*, the avoidance option: referred to as option Y) without being informed of the actual decision by their partner. We instructed the Recipients that the Dictators had not been informed that the Recipients would make the decision. After collecting all data on the Recipients, each Recipient was paid a bonus based on the outcome of the game. Those who chose to receive the money allocated by the Dictator were randomly paired with one of the Dictators and received the money according to their partner's decision. Those who chose the avoidance option received 0.30 USD.

For both roles, participants had to answer questions to check their comprehension of the rules in the experiment before their decision-making. They could not proceed with their decisions until they answered all questions correctly. After submitting their decisions, the Recipients were asked to complete three post-experimental questions. We asked them their expectations of the Dictator's competitive intentions: Recipients predicted the percentage of Dictators who would have chosen the competitive option (hereafter, "expectation") using a slider measure from 0% to 100% in increments of 1% (the initial value was set at 50%). Subsequently, according to methods implemented in previous studies (*Raihani & Bell, 2017*; *Saalfeld et al., 2018*; *Greenburgh, Bell & Raihani, 2019*; *Barnby et al., 2020b*), they also indicated the degree to which Dictators are motivated by harmful intent and self-interest. They rated the extent to which Dictators' decisions were driven by their desire to earn money (hereafter, "self-interest") and by their desire to reduce the Recipient's bonus (hereafter, "harmful intent"). For each motivation, the Recipients indicated their suppositions using a slider ranging from 0 to 100 in increments of one unit (0: not at all, 100: completely, initialized at 50).

## Ethics statements

This research was approved by the Ethics Committee for Human Psychological Research at Teikyo University (No. 627) and was performed per relevant guidelines. Before participating in this study, all participants read a consent form, and informed consent was obtained. They were informed that participation in the study was voluntary and agreed to participate by anonymously checking a box on the consent form page. They could withdraw from participating in the study by closing the web page, even after agreeing to participate.

## Analysis strategy

R version 4.2.2 (*R Core Team, 2020*) was used for all analyses under Mac OS Monterey 12.6.1. The "ggplot2" (*Wickham, 2016*) and "ggdist" (*Kay, 2022*) packages were used for visualization. In the figures, paranoia scores were classified into five levels proposed by *Freeman et al. (2021)* for ease of visualization (1: 0–5; 2: 6–10; 3: 11–17; 4: 18–27; 5: 28–40).

We used regression models when testing pre-registered predictions and conducting unplanned analyses. In the regression models, we added paranoia score, condition (1 = costly, 0 = non-costly), participants' gender (1 = male, 0 = female), and age as predictors. We also added an interaction term between the paranoia score and condition as a predictor. The addition of the interaction term results in a slight deviation from the

pre-registered analysis plans; however, it is intended to conduct more robust analyses to test whether the paranoia score has different effects depending on the conditions. Binary predictors were centered by subtracting from their mean, and continuous predictors were standardized. This coding allowed us to interpret the coefficient of each predictor as the average effect on the outcome when all other predictors were zero (*i.e.*, mean). Thus, the slope of the paranoia score in each model indicates the average effect on the response variable across the two conditions. We also estimated the condition-specific effects of the paranoia score on the response variables from the inferred parameters in the models.

According to previous studies (*Raihani & Bell, 2017*; *Saalfeld et al., 2018*; *Greenburgh, Bell & Raihani, 2019*; *Barnby et al., 2020b*), when we used the expectation, harmful intent, and self-interest scores as response variables, we categorized each score into a five-level ordinal variable (*i.e.*, 0–20, 21–40, 41–60, 61–80, and 81–100) and performed ordinal regression models rather than linear models because of the skewness of the distributions. We also reported the results using linear regression models instead of ordinal regression models in the Supplemental Tables (Tables S7–S9) and confirmed that the conclusions did not change. As in the analyses of other studies (*Raihani & Bell, 2017*; *Saalfeld et al., 2018*; *Greenburgh, Bell & Raihani, 2019*; *Barnby et al., 2020b*), we entered the paranoia score as a continuous predictor in the models, assuming that it would linearly affect the change in the response variable.

We estimated the coefficients of the regression models using Bayesian estimation with Markov Chain Monte Carlo (MCMC) simulations. The Bayesian estimation inferred the posterior distributions (*i.e.*, intervals) of the parameters in the models. For each model, we reported the posterior mean, standard deviation, and 95% credible intervals (CI) of the parameters. If the 95% CI does not overlap zero, it is similar to the significant effect at the 5% level in the statistical null hypothesis test with the frequentist statistics. A "brms" package was used to conduct the Bayesian regression analyses (*Bürkner, 2017*). For the MCMC simulations, we conducted 5,000 iterations per chain, and four chains were performed. For each parameter, we used weakly informative prior distributions to efficiently obtain converged estimates. For the intercept and slopes in the models, the normal distribution with a mean of zero and a standard deviation of 10 was set as a prior distribution. For residual errors in the linear regression models, a half-Cauchy distribution with a shape parameter of zero and a scale parameter of five was set as a prior distribution of standard deviation.

Nine responses of "prefer not to say" for gender were coded as missing values. We used all participants' data for the regression analyses by imputing missing values using the multiple imputation method. Before model fitting was carried out, we imputed the missing values for gender using the "mice" package in R (*van Buuren & Groothuis-Oudshoorn, 2011*). The number of imputations was set to 100.

# RESULTS

## Descriptions

The average paranoia score of the Dictators was 6.01 (standard deviation = 8.43, range = 0–40), whereas that of the Recipients was 6.47 (standard deviation = 8.75,
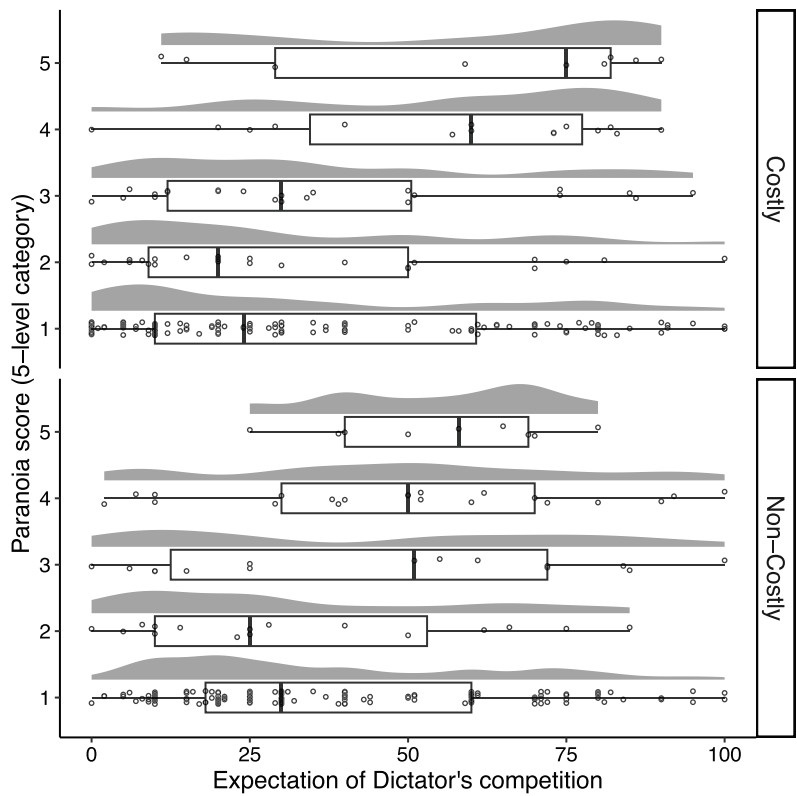

**Figure 2 Distributions of the Recipients' expected percentage of Dictators who would choose the competitive allocation as a function of the Recipients' paranoia levels.** Each point represents each Recipient, and random vertical jitter was added to each point for ease of visibility. The box, the thick line in each box, and the whisker represent the IQR, the median, and the distances 1.5 × IQR, respectively.

range = 0–40). Figure S1 illustrates the distribution of the paranoia scores separately for each role and condition. The difference in paranoia scores between the two roles was statistically insignificant (Wilcoxon's rank-sum test: $p = 0.10$).

We found that 1.65% (4/243) and 3.40% (8/235) of Dictators chose the competitive options in the costly and the non-costly conditions, respectively. We found no significant difference in the frequency of competitors between the two conditions (Fisher's exact test: $p = 0.25$).

Moreover, we found that 25.73% (53/206) and 32.0% (64/200) of Recipients selected the avoidance option in the costly and non-costly conditions, respectively. There was no significant difference in the frequency of Recipients who chose the avoidance option between the two conditions (Fisher's exact test: $p = 0.19$).

## Prediction 1: highly paranoid people are more likely to expect that others will choose a competitive allocation than less paranoid people

After the Recipients completed their decisions, they predicted the percentage of Dictators who they thought would choose the competitive option. Figure 2 illustrates the distribution of the Recipients' expectation scores. On average, the Recipients expected that 36.46%

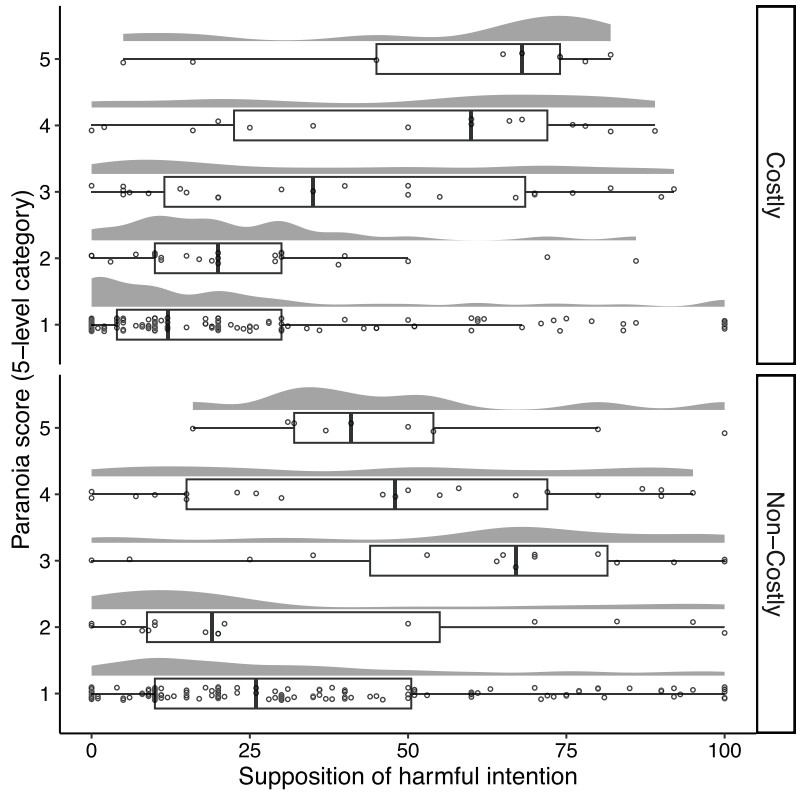

**Figure 3 Distributions of the Recipients' supposition concerning the Dictators' harmful intent as a function of the Recipients' paranoia levels.** Each point represents each Recipient, and random vertical jitter was added to each point for ease of visibility. Boxplots indicate the distributions of the paranoia score. The box, the thick line in each box, and the whisker represent the IQR, the median, and the distances 1.5 × IQR, respectively.

(95% confidence interval [32.25, 40.66]) and 39.88% (95% confidence interval [36.11, 43.65]) of Dictators would choose the competitive option in the costly and the no-costly conditions, respectively. We performed ordinal logistic regression with the expectation score as a response variable and summarized the result in Table S1. The interaction term between the paranoia score and the condition did not predict a change in expectation scores (Table S1: estimate = 0.19, 95% CI [−0.16, 0.54]). We observed a positive effect of paranoia scores on higher expectation scores across conditions (Table S1: estimate = 0.43, 95% CI [0.24, 0.61]). The paranoia score predicted a higher expectation score in each condition (Costly condition: estimate = 0.52, 95% CI [0.24, 0.79]; Non-costly condition: estimate = 0.33, 95% CI [0.10, 0.56]). Hence, this result is consistent with prediction 1.

### Prediction 2: highly paranoid people are more likely to suppose that others will have a harmful intent than less paranoid people

Figure 3 illustrates the distribution of harmful intent scores as a function of paranoia level. The mean of the harmful intent scores were 29.57 (95% confidence interval [25.45, 33.69]) and 37.67 (95% confidence interval [33.39, 41.95]) in the costly and non-costly conditions,

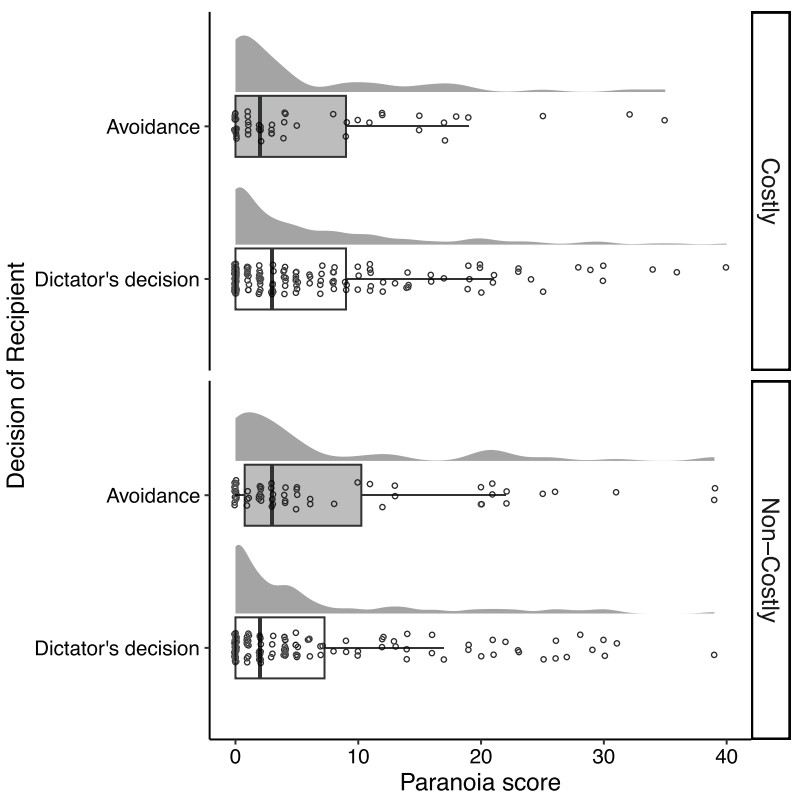

**Figure 4 Distributions of the Recipients' paranoia score according to their decisions in the Dictator Game.** Each point represents each Recipient, and random vertical jitter was added to each point for ease of visibility. The box, the thick line in each box, and the whisker represent the IQR, the median, and the distances 1.5 × IQR, respectively.

respectively. The ordinal regression model using the harmful intent score as a response variable revealed that the interaction term between the paranoia score and the condition did not predict a change in the harmful intent score (Table S2: estimate = 0.25, 95% CI [−0.10, 0.60]). We found an overall positive effect of the paranoia score on the harmful intent score (Table S2: estimate = 0.50, 95% CI [0.32, 0.69]). Recipients with increased paranoid ideation estimated the Dictator's intent as harmful more than those with lower ideation in both the costly (estimate = 0.63, 95% CI [0.36, 0.89]) and non-costly conditions (estimate = 0.38, 95% CI [0.15, 0.62]). Hence, this result is consistent with prediction 2.

### Prediction 3: highly paranoid people are more likely to prefer a reward that they can get for sure to one allocated by others than less paranoid people

Figure 4 illustrates the distribution of Recipients' paranoia scores separately, based on their decisions in the DG. We performed the logistic regression model using the Recipient's decision as a response variable and found no effect for any of the explanatory terms (Table S3). In both conditions, the paranoia score was not associated with the avoidance behavior of Recipients in the Dictator game (Costly condition: estimate = −0.08, 95% CI

[–0.44, 0.25]; Non-costly condition: estimate = 0.11, 95% CI [–0.19, 0.39]). Hence, prediction 3 is not supported.

## Unplanned exploratory analysis: the association between Dictator's competitive choice and paranoia

Figure S2 illustrates the distribution of the Dictators' paranoia scores as a function of their decision in the DG. The logistic regression model using the Dictator's decision as a response variable revealed that the paranoia score was not related to choosing the competitive allocation (Table S4: estimate = –0.11, 95% CI [–1.21, 0.65]).

## Unplanned exploratory analysis: the association between conjecturing Dictator's self-interest and paranoia

Figure S3 illustrates the distribution of the Recipients' self-interest supposition scores as a function of the paranoia level. We performed ordinal logistic regression models using the supposition of the Dictator's self-interest as a response variable. The results showed that only a positive effect of the condition was observed (Table S5: estimate = 0.95, 95% CI [0.57, 1.34]): Recipients in the costly condition assumed that the Dictator's motivation was driven by self-interest more than those in the non-costly condition. Similar to previous findings (*Raihani & Bell, 2017*; *Saalfeld et al., 2018*; *Greenburgh, Bell & Raihani, 2019*; *Barnby et al., 2020b*), we found no overall effect of paranoia score on the supposition of self-interest (Table S5: estimate = –0.08, 95% CI [–0.27, 0.11]). In each condition, the paranoia score did not predict a change in the supposition of self-interest (Costly condition: estimate = –0.15, 95% CI [–0.42, 0.13]; Non-costly condition: estimate = –0.01, 95% CI [–0.26, 0.25]).

## Unplanned exploratory analysis: factors affecting avoidance behavior of Recipients

We performed an exploratory analysis to find the factors affecting the probability of the Recipient choosing the avoidance option using all the obtained variables. We performed a logistic regression using the Recipients' decisions as a response variable. We used the paranoia score, condition, the interaction term between the paranoia score and the condition, gender, age, and post-experimental questionnaire items (*i.e.*, the expectation of Dictators' competitiveness, supposition of harmful intent, and self-interest) as predictors.

Table S6 shows the results. We found that the expectation score had an effect on predicting the avoidance behavior of the Recipient: the more Recipients expected that the Dictators would choose the competitive option, the more they decided to select the avoidance option (Table S6: estimate = 0.72, 95% CI [0.47, 0.98]). Moreover, the harmful intent score was positively associated with avoidance behavior; the more Recipients supposed that the Dictators would have harmful intentions, the more they selected the avoidance option (Table S6: estimate = 0.27, 95% CI [0.02, 0.53]). Other variables (*i.e.*, paranoia, self-interest, condition, gender, and age) did not affect the Recipients' decisions.

## DISCUSSION

We examined whether paranoid thinking was associated with the belief that others were competitors who aimed to maximize the difference in payoffs. In our DG, Dictators choose either fair or competitive allocation, while selecting competitive allocation does not increase their payoffs. The proportion of Dictators who chose competitive allocation was low. Nevertheless, most Recipients estimated a large number of competitors, even in the costly condition in which Dictators must incur costs if they choose the competitive allocation. Moreover, participants with high-level paranoid ideations were more likely to expect others to choose a competitive distribution and assume that others had harmful intentions than those with low-level ideations. The results suggest that overestimation of others' competitive orientation could be observed as a common propensity among the general population. In addition, the current study indicated a strong association between paranoid thinking and the supposition of others' harmful or competitive intentions in uncertain situations.

However, we could not find an effect of paranoid thinking on enhancing behavior to avoid competitive allocations. The exploratory analysis revealed that the expectation of the Dictator's competitiveness and the supposition of the Dictator's harmful intention independently influenced the increase in avoidance behavior. A recent study reported no association between paranoid thinking and betrayal aversion (*Bohnet & Zeckhauser, 2004*): that is, the tendency to avoid risks in paranoia did not differ between social and non-social framing (*Greenburgh et al., 2021*). The behavior measured in the present study may be similar to the concept of betrayal aversion in terms of the propensity to avoid interpersonal risk. These findings seem contrary to theoretical views suggesting a link between paranoia and safety behaviors such as avoidance (*Freeman et al., 2007*; *Freeman, 2016*).

However, previous findings have indicated a relationship between paranoia and reactions to perceived threats. Paranoid thinking is associated with defection or distrust in economic games (*Fett et al., 2012*; *Ellett et al., 2013*). Paranoid people engage in punitive behavior toward unfair opponents (*Raihani & Bell, 2018*) (although similar findings were not observed in *Raihani et al. (2021)*). Moreover, a weak association between paranoid thinking and a preemptive strike has been observed (*Horita, 2021*). Overall, these findings suggest that both the subjective perception of threat and an opportunity to affect the payoffs of the other, whom the paranoid believes to have harmful intentions, may work as conditions for whether paranoia responds to perceived threats. It may be necessary for paranoid people to reciprocate, rather than avoid, others they believe to be malicious. As discussed below, paranoid ideation does not reflect the propensity to attack others unilaterally. To explain the response to threats in paranoia, we should consider two conditions: the belief that others would have a harmful intent, and that others have a means of threatening.

The lack of an association between avoidance and paranoid thinking in this study may also be attributed to other confounded variables. For example, the relationship between paranoia and the pursuit of self-interest in economic games (*i.e.*, payoff-maximizing) has been highlighted (*Raihani & Bell, 2018*). A previous study examined the association

between paranoid thinking and self-reported threat avoidance in daily life (*Freeman et al., 2007*). In contrast, the present study used an economic experiment based on a game-theoretical paradigm, in which there was a trade-off between avoiding threats and losing the opportunity to gain more benefits. Even though individuals with higher paranoia levels presumed that others had competitive preferences, they might have been motivated to take risks (*i.e.*, receive the money according to the Dictators' decision) to earn as much money as possible. The motivation to avoid perceived threats may counteract the desire to earn more payoffs. Therefore, this study may have found no association between paranoia and avoidance behavior. Other psychological variables reflecting a self-interested orientation, such as the participant's own SVO, should be considered control variables.

We also explored the link between paranoia and antisocial behavior (*i.e.*, reducing the other's interests) by observing the Dictator's behavior. A previous study showed that people rarely engage in a preemptive strike when they can unilaterally reduce others' payoffs (*Simunovic, Mifune & Yamagishi, 2013*). People whose SVOs were classified as competitors were rarely observed (*Au & Kwong, 2004*). We also found that few Dictators chose competitive allocation, and that paranoid thinking did not affect the probability of selecting competitive allocation, regardless of the cost of choosing it. Previous studies have pointed to a relationship between paranoia and selfish or antisocial behaviors. Individuals with more paranoid thoughts show low generosity in economic games (*Fett et al., 2012*; *Ellett et al., 2013*; *Raihani & Bell, 2018*; *Savulich et al., 2018*). Other studies have argued that paranoia is related to self-reported violence (*Coid et al., 2016*) and a propensity to enjoy negative social interactions (*Raihani et al., 2021*). However, we confirmed that paranoia did not lead to antisocial behavior in an experimental situation in which individuals could unilaterally attack their opponent without concern for others' harm. From previous and current research, it is possible that perceiving the harmful intentions of others is a prerequisite for paranoia to induce antisocial behaviors toward the expected harm.

We found no interaction between the paranoia scores and the conditions of the Dictator's costs in selecting a competitive allocation. On a similar note, previous studies using economic games have confirmed the lack of interaction between pre-measured paranoia and experimental manipulations; participants with a higher level of paranoia are more likely to attribute others' intentions as harmful than those with a lower level, irrespective of experimental conditions (*Saalfeld et al., 2018*; *Greenburgh, Bell & Raihani, 2019*). Previous studies have concluded that pre-existing paranoid thinking reflects a lower threshold for detecting social threats. In our experiments, participants with higher paranoia scores were more likely to suppose the Dictator's competitive or harmful intentions, even when there were no reasons for rational individuals to choose competitive allocations (*i.e.*, in the costly condition). This study also strengthens the argument that pre-existing paranoid thinking represents a default lower threshold for detecting social threats, rather than strong reactivity or insensitivity to social contexts.

Avoiding perceived threats leads to a repetitive cycle of persistent paranoid beliefs by preventing the processing of evidence that their belief is without foundation (*Freeman et al., 2007*; *Freeman, 2016*). Exploring interventions to reduce paranoid delusions is an

interest in research on paranoia (*Freeman, 2016*; *Monaghesh, Samad-Soltani & Farhang, 2022*). The current study suggested that there was a substantial difference between the actual proportions of competitors and their beliefs in people's minds. Knowledge about the low proportion of competitors in the general population may contribute to reducing negative beliefs about others and increasing confidence in safety. Another prior study has provided experimental results that experiences in repetitive interactions with fair partners reduced harmful intention attribution (*Barnby et al., 2020b*). Investigating how learning others' preferences contributes to the reduction of paranoid thinking would provide insights into clinical issues as well as an understanding of the cognitive mechanisms of paranoia.

This study has some limitations. First, other potential mental health factors associated with paranoia should be controlled. Several mental health problems, such as depression, worry, and anxiety, are associated with paranoid thinking (*Freeman et al., 2011*). Furthermore, these factors may be related to responses to perceived threats in paranoia. For example, paranoid individuals who believe that they deserve to be persecuted are likely to have high levels of depression or low self-esteem (*Chadwick et al., 2005*; *Melo, Taylor & Bentall, 2006*): therefore, this type of paranoia may tend to be submissive to social threats. It would be necessary to control for these personal factors by assessing mental health to more rigorously examine the association between paranoia and responses to social threats. Second, there is room for improvement in the experimental settings. This study was conducted with a small stake size. Participants might have been more likely to engage in risk-taking behavior than to avoid the risk because they lost only a small amount of money, even if their opponent chose competitive allocation. In addition, although we recruited participants from MTurk workers in the U.S., we should examine the validity of the findings with other samples. Participants recruited online may be more representative than student samples, but a more culturally diverse sample should be targeted to examine general psychological foundations. Follow-up studies in different experimental settings are needed to verify the robustness of these findings.

As previously mentioned, the need to measure SVOs of self was one of the limitations of the current study. A recent study showed the possibility that paranoia is associated with one's own SVOs and predictions of others' SVOs (*Barnby, Raihani & Dayan, 2022*). *Barnby, Raihani & Dayan (2022)* conducted an experimental task in which participants predicted others' social preferences and found that a process of Bayesian inference better explained the participants' cognitive process in the experiment; they used their own preferences as a prior belief about others' preferences and updated it *via* experiences. In addition, *Barnby, Raihani & Dayan (2022)* revealed that paranoia is associated with preferences for earning more than a partner and less flexible updating of a belief about others' preferences. Overestimation of others' competitiveness in paranoia may be based on these cognitive tendencies; paranoid individuals project their own selfish preferences onto others and have difficulty changing their beliefs. However, this study also proposed that paranoia was not associated with choosing competitive allocations. Further examination of the association between SVOs and paranoia is necessary for understanding the effects of paranoia on cognition and behavior in social interactions.

Paranoia has been observed to have a continuous propensity in a general non-clinical sample. The findings of this study also suggest that the general belief that others have a competitive orientation can be observed. Recently, the evolutionary foundation of paranoia as a normal psychological function has been discussed (*Green & Phillips, 2004*; *Raihani & Bell, 2019*; *Bell, Raihani & Wilkinson, 2021*). Although excessive paranoid delusions are associated with mental disorders, such as schizophrenia, mild paranoid thinking may play an important role in detecting and responding to social threats. Aggression (*Coid et al., 2016*; *Horita, 2021*) and submission (*Freeman et al., 2005*; *Gilbert et al., 2005*) are strategies against social threats. The contribution of the current study is that it provides empirical evidence of no connection between paranoia and avoidance behavior toward social threats in social interactions. Further research examining the repertoire of strategies for coping with social threats in paranoia would be useful for understanding the evolutionary foundations of paranoia.

## CONCLUSIONS

In this study, we examined whether paranoid thinking was associated with the supposition that others have competitive intentions using a modified DG. Although the Dictators who chose the competitive allocation option were rarely observed, some Recipients overestimated the other's competitive intentions and avoided the Dictator's allocations. Recipients with high-level paranoid thinking were more likely to anticipate that others would choose a competitive allocation and have harmful intentions, even if selecting the competitive option was costly for the Dictators. However, paranoid thinking did not strongly affect behavior to avoid competitive allocation. These results suggest that paranoid thinking acts as a function of detecting others' competitive intentions, although the avoidance of perceived threats is not a general strategy in paranoia.

## ACKNOWLEDGEMENTS

We would acknowledge Editage for the English language editing of the first manuscript.

### Funding

This research was supported by Grants-in-Aid for Scientific Research from the Japan Society for the Promotion of Science (Grant No. JP18K13276 and JP21K02984).
The funders had no role in study design, data collection and analysis, decision to publish, or preparation of the manuscript.

### Grant Disclosures

The following grant information was disclosed by the authors:
Grants-in-Aid for Scientific Research from the Japan Society for the Promotion of Science: JP18K13276 and JP21K02984.

### Competing Interests

The authors declare that they have no competing interests.

## Author Contributions

- Yutaka Horita conceived and designed the experiments, performed the experiments, analyzed the data, prepared figures and/or tables, authored or reviewed drafts of the article, and approved the final draft.

## Ethics

The following information was supplied relating to ethical approvals (*i.e.*, approving body and any reference numbers):

The Ethics Committee for Human Psychological Research at Teikyo University.

## Data Availability

The data, code for analyses, and Supplemental Information are available at Open Science Framework: Horita, Yutaka. 2023. "Paranoid Thinking and Perceived Competitive Intention." OSF. February 16. DOI 10.17605/OSF.IO/YKXDW.

## Supplemental Information

Supplemental information for this article can be found online at http://dx.doi.org/10.7717/peerj.15003#supplemental-information.

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
