# Peer review of "Paranoid thinking and perceived competitive intention"

_PeerJ, doi:10.7717/peerj.15003_

## Round 0.1 · original submission · Minor Revisions

I have now received the reviewers' comments on your manuscript. They have suggested some minor revisions to your manuscript. Therefore, I invite you to respond to the reviewers' comments and revise your manuscript.

Reviewer 1 ·

Basic reporting

Thankyou to the author for a well-controlled and interesting paper following up the tendency of those with high paranoia to assume their partner is competitive. The study follows on well from prior work in the field, and care has been taken to ensure predictions are tested rigorously. It is found that paranoia has an overall effect to expect more competitive decisions, irrespective of the cost of these decisions, and that paranoia exaggerates harmful intent. Paranoia was not related to avoidance behaviour. I have a few minor comments and suggestions, but I think the paper is mostly very strong as it stands.

The introduction is laid out well, and the methods are very clear, including the analysis packages used to assess the data, and where plans deviated from the preregistration.

Experimental design

The experimental design makes sense and the author allocates participants reasonably to each condition on a random schedule.

Validity of the findings

The findings are strong and are carefully noted to either address preregistered hypotheses or contribute to exploratory analysis. There may be some questions about the validity of the use of mTurk, but it appears that the authors took care to select participants that were indeed not bots or workers that automate their games. Finally, did the authors enter paranoia into the regression models as a continuous variable or as an ordinal variable?

Additional comments

I would have liked to see some discussion around the lack of interaction between paranoia and condition. While there is a main effect of paranoia on competitive expectations and harmful intent, there is no apparent sensitivity to the cost of this for Partner 2. Is it that paranoia is insensitive to social context? Prior theoretical work (Raihani & Bell, 2019; Nat Hum Behav.) would suggest that as paranoia gets more severe, inferences become less tethered to the social environment, however paranoia in your sample was relatively mild. How do you square this lack of interaction with prior theory?

Likewise, theories of avoidance in paranoia (Freeman et al., 2016; Lancet) suggest that severe paranoia may engender avoidance behaviours leading to a repetitive cycle. Are you suggesting with your results that mild paranoia may not display these traits, and that more severe paranoia only tends to manifest avoidance? Could it be that other cognitive biases are involved with respect to avoidance that are not measured here?

Finally, in the limitations section, it is noted that the design may want to consider the SVO of the participant. Prior work may be able to speak to this (Barnby et al., 2022; Cognition); it would support your findings and suggest that those with high paranoia are less prosocial, and these priors form the basis of their rigid beliefs that others are more competitive.

·

Basic reporting

In the present study the Author examined whether paranoia reflects the irrational belief that others have a competitive intention and is associated with avoiding perceived competition.
Overall, I found this study timely, original, well-conducted and scientifically sound: I enjoyed reading it. However, I have some minor suggestions aimed at improving the quality of the paper, and these are outlined below:
1) In the introduction, a brief note on the fact that paranoia might be a multidimensional disorder which includes several subtypes with different neurobiological underpinnings, should be added.
2) Were the participants consecutive or randomly selected? And how many subjects were screened, but refused to participate or excluded and why?
3) Was also the presence of an intellectual disability evaluated, how, and used as an exclusion criterion?
4) Nothing changes to the worth of the study, in my opinion. Still, I guess why the Authors decided not to use other controlling rating scales as PHQ-9 or others to strengthen the results, controlling for potential confounding variables. Maybe it should be added to the limitations.
5) I suggest slightly improving the English language with the help of a fluent speaker.

Experimental design

Please, see above

Validity of the findings

Please, see above

Additional comments

Please, see above

---

## Round 0.2 · accepted · Accept

In my opinion, this manuscript has been revised with attention to the reviewers' comments and can now be published. However, one of the reviewers has provided suggestions that you can apply in galley proof if you wish.

Reviewer 1 ·

Basic reporting

I am happy with the current state of the article.

Experimental design

I am happy with the current state of the article.

Validity of the findings

I am happy with the current state of the article.

Additional comments

Thank you very much for revising the manuscript.

There is one section in the discussion that, on reflection, I feel slightly embarrassed about concerning my use of words. My comment with respect to the SVO of the participant was not meant to be an exercise of self-citation; I had included a previous example of our work as a reference but by no means expected the citation to be included in the MS for it to be accepted. I had not intended for that to be the interpretation. I would be very happy for the author to remove the citation if they did not feel it adds to the discussion, or, if they do feel it does add to the discussion, I might ask that they also consider including the following references (listed below) for a firmer grounding of the claim. This is not a necessary aspect for publication (I do not need to review the MS again) and is more of a suggestion in light of my slightly clumsy wording:

Tarantola, T., Kumaran, D., Dayan, P., & De Martino, B. (2017). Prior preferences beneficially influence social and non-social learning. Nature Communications, 8(1), 817.

Krueger, J., & Clement, R. W. (1994). The truly false consensus effect: an ineradicable and egocentric bias in social perception. Journal of personality and social psychology, 67(4), 596.

Andersen, S. M., & Chen, S. (2002). The relational self: an interpersonal social-cognitive theory. Psychological review, 109(4), 619.

Overall the manuscript operates a very careful methodology and I appreciate the attention to detail of the reviewer. I look forward to seeing the manuscript in print.

I identify myself as Joseph Barnby.

·

Basic reporting

I believe that this paper is very interesting and worthy of publication

Experimental design

Please, see above

Validity of the findings

Please, see above